# SHIFTING ATTENTION TO RELEVANCE: TOWARDS THE UNCERTAINTY ESTIMATION OF LARGE LANGUAGE MODELS

## ABSTRACT

While Large Language Models (LLMs) have demonstrated remarkable potential in natural language generation and instruction following, a persistent challenge lies in their susceptibility to "hallucinations", which erodes trust in their outputs. Although Uncertainty Quantification (UQ) presents a promising solution, its accurate implementation within the context of LLMs remains a significant hurdle. To address this critical roadblock, our research originates from a fundamental heuristic insight: tokens within auto-regressive LLM-generated text do not equally reflect the underlying meaning. Some tokens carry greater relevance and representativeness than others, owing to the phenomenon of "linguistic redundancy", wherein a select few keywords suffice to convey the essence of lengthy sentences. Regrettably, existing methodologies treat all tokens with equal importance when estimating uncertainty, disregarding these inherent generative inequalities. Our analysis reveals a significant issue with state-of-the-art: numerous tokens (and sentences) of limited semantic significance receive equal or even excessive weighting during uncertainty estimation. To rectify this bias, we propose to jointly **S**hifting **A**ttention to more **R**elevant (*SAR*) components, at both the token- and the sentence-levels for accurate uncertainty estimation. We conduct extensive experiments involving a range of popular "off-the-shelf" LLMs, including instruction-tuned LLMs such as Vicuna, WizardLM, and LLaMA-2-chat, as well as pretrained LLMs like OPT and LLaMA, with model sizes extending up to 33B parameters. We carry out evaluation across various free-form question-answering tasks, encompassing domains such as reading comprehension, science Q&A, and medical Q&A. Our experimental results, coupled with a comprehensive demographic analysis, demonstrate the superior performance of SAR in addressing the challenges of uncertainty estimation within the realm of LLMs.

## 1 INTRODUCTION

Large Language Models (LLMs) have shown remarkable capabilities in intent understanding He & Garner (2023), multi-round conversation Long (2023); Chen et al. (2023), logical reasoning Creswell et al. (2022); Pan et al. (2023), and also disclose great potential in scientific discovery Birhane et al. (2023). For instance, the recent ChatGPT, BARD, GPT-4, pre-trained on large-scale corpora and carefully aligned to human preferences Christiano et al. (2017); Ouyang et al. (2022), profoundly shape the range of what AIs could do, and how they communicate with humans.

Despite the surprising progress, LLMs are proven to be vulnerable to widely known reliability issues, such as hallucination Manakul et al. (2023a) and factual errors Bian et al. (2023); Karpinska & Iyyer (2023); Gekhman et al. (2023). Uncertainty estimation is one of the most popular approaches to answering when humans can trust the generations of LLMs, which is critical for Human-AI interaction applications (e.g., therapy and mental health Lin et al. (2023); Sharma et al. (2023)) where humans need to densely communicate with LLMs. In these applications, the resulting behaviors will be largely affected by the generations from LLMs.

Unfortunately, uncertainty estimation still remains challenging due to various uncertainty sources (e.g., aleatoric uncertainty and epistemic uncertainty Kendall & Gal (2017)).

Especially for free-form language models where the model complexity is high and the solution domain is effectively unbounded, i.e., any generation that has the same semantic as the ground-truth answer should be deemed as correct, the uncertainty estimation problem is significantly different from the well-studied classification models or any other models that have specific labels.

Prior works in this direction estimate uncertainty by prompting LLMs to answer confidence Lin et al. (2022a); Kadavath et al. (2022a) or designing logits- or entropy-based measurements Malinin & Gales (2021; 2020); Kuhn et al. (2023). The most recent work proposes *Semantic Entropy* (*SE*) Kuhn et al. (2023) where generations sharing the same meaning (or semantic equivalence sentences) are gathered in a semantic cluster. Then the cluster-wise entropy is calculated as the uncertainty measurement.

Our motivation is derived from an intuitive fact: *tokens are created unequally in presenting semantics*. Namely, some tokens (e.g., nouns, verbs) are more meaningful than other tokens (e.g. definite articles). For example, for a given question "*What is the ratio of the mass of an object to its volume?*" and a model generation "*density of an object*". It is clear that "'density" is the most relevant token in presenting semantics than the rest tokens. We term the former as *relevant tokens* and the rest tokens as *irrelevant tokens*. Prior works treat each token equally when estimating uncertainty, which is counter-intuitive ( Figure 1). Therefore, we ask:

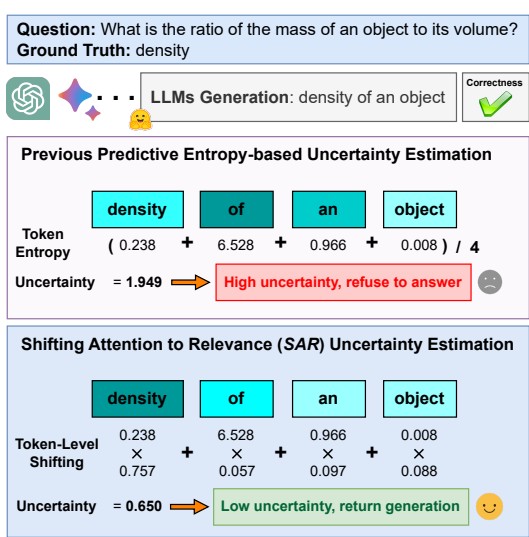

Figure 1: Irrelevant tokens (or sentences) may commit majority uncertainty in free-form generations, such as the token "of" committing extremely large uncertainty misleads the uncertainty estimation of LLMs. We term these observations as generative inequalities and tackle them by shifting attention to more relevant components.

*Are relevant tokens more critical than irrelevant tokens when estimating uncertainty?*

To answer this question, we first investigate how token-level generative inequality affects uncertainty estimation in LLMs. Specifically, we first measure the *relevance score* of each token by comparing the semantic change before and after removing this token from the generation. A larger semantic change means more relevance for this token and vice versa. Then we quantify the *uncertainty proportions*, i.e., the uncertainty committed by this token. At last, we analyze the correlation between relevance and uncertainty proportion. Our results reveal that there are large amounts of tokens containing very limited semantics yet are weighted equally or even heavily when evaluating uncertainty. We further generalize to the sentence-level inequality by assessing relevant sentences and irrelevant sentences where similar observations are observed.

Based on these observations, we propose a simple attention-shifting method, by jointly examining the relevance of each component and reassigning its attention, from both the token level and the sentence level, termed as **S**hifting **A**ttention to **R**elevance (*SAR*). *SAR* is evaluated on multiple popular open-source instruction-tuned LLMs (e.g., Vicuna Zheng et al. (2023), LLaMA-2-chat Touvron et al. (2023b), WizardLM Xu et al. (2023)), with model size up to 33B, and popular pre-trained LLMs (e.g., OPT Zhang et al. (2022), LLaMA Touvron et al. (2023a)) with model sizes up to 30b, over cross-domain free-form question-answering tasks, such as the conventional NLP domain (e.g., CoQA Reddy et al. (2019), TriviaQA Joshi et al. (2017) and SciQ Welbl et al. (2017)) and medical domain (e.g., MedQA Jin et al. (2020), MedMCQA Pal et al. (2022)). Experimental results demonstrate *SAR*'s superior performance. Our contributions can be summarized as the following:

- We disclose that uncertainty estimation is significantly affected by token- and sentence-level generative inequality, i.e., irrelevant tokens or sentences might be over-valued when estimating uncertainty.

- We mitigate the two inequality biases by **S**hifting **A**ttention to **R**elevance (*SAR*), which jointly examines the relevance of each token and sentence, and reassigns attention when estimating uncertainty.
- We conduct experiments over "off-the-shelf" instruction-tuned LLMs and popular pre-trained LLMs, across various free-form question-answering tasks. Experimental results demonstrate that *SAR* outperforms previous state-of-the-art by a large margin.

## 2 RELATED WORKS

**Uncertainty Estimation in Conventional NLP Tasks.**    Uncertainty Estimation of machine translation (MT) has been studied for years to evaluate the performance of MT better. Ott et al. (2018) access uncertainty by comparing multiple model outputs to multiple references with inter-sentence BLEU. Glushkova et al. (2021) measure uncertainty through techniques of Monte Carlo dropout Gal & Ghahramani (2016) and deep ensembles Lakshminarayanan et al. (2017). Fomicheva et al. (2020) use uncertainty quantification methods to improve probability estimates in neural networks for better quality estimation. Lahlou et al. (2021) proposed Direct Epistemic Uncertainty Prediction, a model-agnostic framework, for estimating epistemic uncertainty in machine learning models. For regression tasks, Wang et al. (2022) use uncertainty estimation to address both data uncertainty and model uncertainty, and Malinin et al. (2020) proposes a method for uncertainty estimation using Prior Networks to obtain interpretable measures of uncertainty at a low computational cost. For Natural Language Understanding tasks, Talman et al. (2023) use uncertainty estimation by applying Bayesian uncertainty modeling using Stochastic Weight Averaging-Gaussian.

**Uncertainty Estimation in LLMs.**    Although uncertainty estimation has been thoroughly examined in models with distinct labels, such as classification models Ulmer et al. (2022); Vazhentsev et al. (2022), it is still under-explored for popular free-form LLMs, e.g., GPT Radford et al. (2019), OPT Zhang et al. (2022), LLaMA Touvron et al. (2023a). These models present a unique challenge in uncertainty estimation as their solution domains are flexible and effectively infinite, i.e., any generation can be deemed correct as long as the semantics align consistently with the real answer.

Xiao et al. (2022) conducts large-scale empirical evaluations on how the configuration (e.g., model size, architecture, training loss) of LLMs affect uncertainty. Lin et al. (2022a); Kadavath et al. (2022a) propose to quantify uncertainty by directly prompting the language models to answer the uncertainty with respect to their generations. Manakul et al. (2023b) measures the faithfulness of generations by quantifying the consistency of generations, i.e., generations should be consistent if the model really captured the concept. Malinin & Gales (2021) examines the uncertainty of free-form LLMs by calculating the accumulative predictive entropies over multiple generations. Recently, Semantic Entropy (*SE*) Kuhn et al. (2023) is presented to tackle the "semantic equivalence" difficulty in uncertainty quantification. *SE* gathers generations sharing the same semantics into clusters and performs cluster-wise predictive entropy as the uncertainty measurement.

We aim to design metrics from multiple generations to characterize the uncertainty of LLMs. Our work focuses on the token- and sentence-level generative inequalities, which are not explored by prior works in uncertainty estimation.

## 3 GENERATIVE INEQUALITY IN UNCERTAINTY ESTIMATION

Tokens are created unequally in reflecting the meaning of the generation yet they are treated equally when estimating uncertainty. We term these inequalities as *generative inequalities* and investigate how they affect uncertainty estimation.

### 3.1 PRELIMINARIES

LLMs normally generate sentences in a free-form and auto-regressive manner, i.e., progressively predicting the probability distribution of the next token. We denote by $x$ the input (or the prompt) and $s$ the generated sentence with the length of $N$. Then, for a given LLM, the probability of generating $z_i$ as the $i$-th token can be described as $p(z_i|s_{<i}, x)(1 \leq i \leq N)$, where $s_{<i}$ refers to the previously generated tokens $\{z_1, ..., z_{i-1}\}$.

**Baseline.** We use the popular Predictive Entropy (*PE*), described in Kadavath et al. (2022b), as the baseline and investigate how it is affected by generative inequalities in this section. The Predictive Entropy (*PE*) is defined as the entropy over the whole sentence $s$:

$$PE(\boldsymbol{s}, \boldsymbol{x}) = -\log p(\boldsymbol{s}|\boldsymbol{x}) = \sum_i -\log p(z_i|\boldsymbol{s}_{<i}, \boldsymbol{x}). \tag{1}$$

It can be interpreted as the accumulation of the token-wise entropy.

## 3.2 TOKEN-LEVEL GENERATIVE INEQUALITY

As mentioned before, generative inequality refers to an observation where some tokens contain limited semantics yet are equally valued when estimating the uncertainty of a generation, which is counter-intuitive. To outline this observation, we specify two quantities for each token: how much semantics the token contains, i.e., the *relevance*, and how much uncertainty the token committed, i.e., the *uncertainty proportion*.

For a given prompt $\boldsymbol{x}$ and the generated sentence $\boldsymbol{s}$ consisting of $N$ tokens, i.e., $\boldsymbol{s} = \{z_1, z_2, ..., z_N\}$, we quantify the relevance and uncertainty proportion of token $z_i$:

**Relevance.** To measure how important $z_i$ is in reflecting the semantics of $\boldsymbol{s}$, we compare the semantic change before and after removing this token:

$$R_T(z_i, \boldsymbol{s}, \boldsymbol{x}) = 1 - |g(\boldsymbol{x} \cup \boldsymbol{s}, \boldsymbol{x} \cup \boldsymbol{s} \setminus \{z_i\})|, \tag{2}$$

where $g(\cdot, \cdot)$, calculating sentence similarity on a scale of 0 to 1, can be any semantic similarity measurement. In our experiments, we leverage the Cross-Encoder Reimers & Gurevych (2019b)-RoBERTa-large Liu et al. (2019) as this measurement since it is one of the most powerful sentence similarity evaluation models provided by the popular SentenceTransformers Library Reimers & Gurevych (2019a). Generally, larger $R_T(z_i, \boldsymbol{s}, \boldsymbol{x})$ means removing $z_i$ will lead to significant semantic changing, which indicates the importance of $z_i$ and vice versa.

**Uncertainty Proportion.** To measure the proportion of uncertainty committed by $z_i$, we simply derive the ratio from Eq. (1):

$$UP_T(z_i, \boldsymbol{s}, \boldsymbol{x}) = \frac{-\log p(z_i|\boldsymbol{s}_{<i}, \boldsymbol{x})}{PE(\boldsymbol{s}, \boldsymbol{x})}. \tag{3}$$

Larger $UP_T(z_i, \boldsymbol{s}, \boldsymbol{x})$ means $z_i$ commits more uncertainty when estimating the uncertainty of sentence $\boldsymbol{s}$; vice versa.

## 3.3 SENTENCE-LEVEL GENERATIVE INEQUALITY

It has been widely shown that involving multiple generations benefits estimating uncertainty Kadavath et al. (2022b). For instance, *PE* will usually be the arithmetic mean of multiple sentences in practice, i.e., $\frac{1}{K} \sum_k PE(\boldsymbol{s}_k, \boldsymbol{x}) \, (1 \le k \le K)$ where $S = \{\boldsymbol{s}_1, \boldsymbol{s}_2, ..., \boldsymbol{s}_K\}$ consisting of $K$ generated sentences regarding $\boldsymbol{x}$ and $\boldsymbol{s}_k \in S$ is the $k$-th sentence. Therefore it is necessary to study sentence-level generative inequality. Following Section 3.2, for a given sentence $\boldsymbol{s}_i$, we define the sentence-level relevance of $\boldsymbol{s}_i$ as the probability-weighted semantic similarity with other sentences.

$$R_S(\boldsymbol{s}_i, S, \boldsymbol{x}) = \sum_{j=1, j \neq i} g(\boldsymbol{s}_i, \boldsymbol{s}_j) p(\boldsymbol{s}_j|\boldsymbol{x}), \tag{4}$$

where $1 \le i, j \le K$ and $p(\boldsymbol{s}_j|\boldsymbol{x})$ is the generative probability of $\boldsymbol{s}_j$. It is out of an intuitive assumption that sentences are more convincing if they are semantically consistent with other generations. Namely, a sentence that is semantically close to other generations is considered more representative. Besides, the generative probability $p(\boldsymbol{s}_j, \boldsymbol{x})$ provides more confidence for sentence $\boldsymbol{s}_j$ as measuring relevance, i.e., higher $p(\boldsymbol{s}_j, \boldsymbol{x})$ makes $\boldsymbol{s}_j$ more compelling.

Similar to the token-level situation, the sentence-level uncertainty proportion of $\boldsymbol{s}_i$ is defined as:

$$UP_S(\boldsymbol{s}_i, S, \boldsymbol{x}) = \frac{PE(\boldsymbol{s}_i, \boldsymbol{x})}{\sum_k PE(\boldsymbol{s}_k, \boldsymbol{x})}, \tag{5}$$

where $1 \le k \le K$. It is the proportion of uncertainty committed by $\boldsymbol{s}_i$,

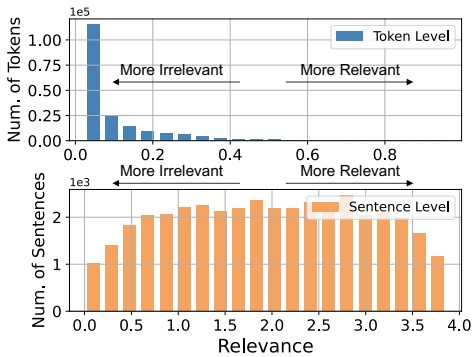

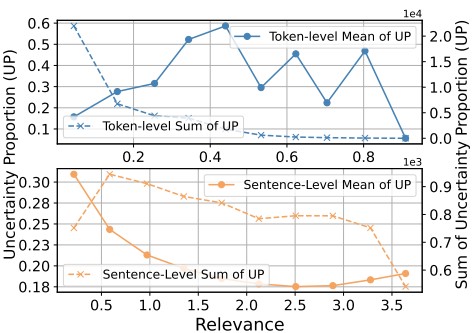

Figure 2: Distributions of relevance scores in both token-level and sentence-level situations. It is shown that there are considerable irrelevant tokens and sentences that appear over generations, especially for the token situations where most tokens are irrelevant.

Figure 3: Correlations between relevance scores and uncertainty proportions in both token-level and sentence-level situations. Irrelevant tokens dominate the total volume of uncertainty estimation. Irrelevant sentences dominate both total volume and average.

### 3.4 ANALYTICAL INSIGHTS

We will leverage the defined *relevance* and *uncertainty proportion* to characterize the generative inequality observations in this section. We utilize CoQA as the dataset and OPT-13b as the model to be examined. For each prompt in CoQA, we will generate 10 sentences, i.e., $K = 10$ in Eq. (4) and Eq. (5). More details of generation can be found in Appendix A.

We first quantify the distributions of token-level relevance scores and sentence-level relevance scores. Results are summarized in Figure 2. For token-level relevance, it is clear that most of the tokens are irrelevant tokens, i.e., they have low relevance scores. It indicates that linguistic redundancy exists widely. In terms of the sentence-level situation, although the distribution is flatter than the token-level situation, the irrelevant sentences still take a considerable amount of all sentences.

We further investigate the correlations between relevance and uncertainty proportions, i.e., how much uncertainty is committed by tokens and sentences with various relevance scores. We calculate these quantities by first independently gathering tokens and sentences into 10 bins with uniform relevance ranges and then averaging the uncertainty proportions of tokens/sentences contained in the same bin. Results are summarized in Figure 3.

For the token-level situation, although tokens with large relevance scores commit slightly higher uncertainty on average, due to a large number of irrelevant tokens, the irrelevant tokens still dominate uncertainty estimation from the perspective of total volume (the dashed line). For the sentence-level situation, it is clear that irrelevant sentences commit more uncertainty than relevant sentences regardless of the average or the total.

These observations demonstrate the existence of generation inequalities and also the uncertainty estimation is highly affected by these inequalities.

## 4 SHIFTING ATTENTION TO RELEVANCE

A natural hypothesis derived from Section 3.4 is that shifting the attention to those relevant components may benefit uncertainty estimation. In this section, we introduce the proposed Shifting Attention to Relevance (*SAR*) in detail.

### 4.1 NOTATIONS

We reuse the notations defined in Section 3.1 where we denote by $x$ the prompt and $S$ the generated $K$ sentences. There will be $N_j$ tokens for each sentence $s_j \in S \, (1 \le j \le K)$.

## 4.2 RELEVANCE DISCOVERY AND SHIFTING

*SAR* corrects generative inequalities by reviewing the relevance of each token and/or sentence and emphasizing uncertainty estimation attention to those more relevant components. Here we introduce token-level shifted measurement and sentence-level shifted measurements:

**Token-Level Shifting.** For a generation $s_j$ regarding prompt $x$, $s_j = \{z_1, z_2, ..., z_{N_j}\}$ contains $N_j$ tokens. We first calculate the normalized relevance score for each token $z_i$ ($1 \leq i \leq N_j$) based on Eq. (2), i.e., $R_T(z_i, s_j, x)$:

$$\tilde{R}_T(z_i, s_j, x) = \frac{R_T(z_i, s_j, x)}{\sum_n^{N_j} R_T(z_n, s_j, x)} \tag{6}$$

Then we enlarge the uncertainty proportions of relevant tokens by re-weighting token entropy according to their normalized relevance scores:

$$E_T(z_i, s_j, x) = -\log p(z_i | s_{<i}, x) \tilde{R}_T(z_i, s_j, x). \tag{7}$$

The token-level shifted predictive entropy defined over $s_j$ can be formulated as:

$$\text{TOKEN}SAR(s_j, x) = \sum_i^{N_j} E_T(z_i, s_j, x). \tag{8}$$

The reason we normalize relevance score in Eq. (6) is two-fold: a) to make tokens comparable across sentences; b) to mitigate the bias posed by sentence length, like the length normalization in Length-normalized Predictive Entropy (*LN-PE*) Malinin & Gales (2020). In this way, the uncertainty proportions of tokens containing strong relevance will be enlarged when estimating uncertainty.

**Sentence-Level Shifting.** As mentioned in Section 3.3, sentences that have higher relevance scores, i.e., semantically consistent, are more convincing than others. Therefore, we simply reduce sentence uncertainty by enlarging sentence generative probability with a relevance-controlled quantity:

$$E_S(s_j, S, x) = -\log(p(s_j | x) + \frac{1}{t} R_S(s_j, S, x))$$
$$= -\log(p(s_j | x) + \underbrace{\frac{\sum_{k \neq j} g(s_j, s_k) p(s_k | x)}{t}}_{\text{sentence relevance}}), \tag{9}$$

where $p(s_j | x) = \prod_i p(z_i | s_{<i}, x)$ is the generative probability of $s_j$ and $t$ is the temperature used to control the scale of shifting. Then, the sentence-level shifted predictive entropy over $K$ sentences can be formulated as:

$$\text{SENT}SAR(S, x) = \frac{1}{K} \sum_k E_S(s_k, S, x). \tag{10}$$

Note that Eq. (9) shares a similar form with *SE* Kuhn et al. (2023), i.e., reducing the uncertainty of semantically consistent sentences. Differently, *SE* achieves this with bi-directional entailment prediction and we achieve this with weighted relevance scores. With manual examination, we found that around 36.7% of the entailment predictions are undesirable, over the long generations that have more than 20 tokens on average (120 questions in total). Instead, our sentSAR leverages the more "soft" sentence similarity to calculate the relevance score, which is more desirable for long and complex sentences.

## 4.3 OVERALL MEASUREMENT

Token-level shifting and sentence-level shifting are conceptually different as they emphasize different perspectives of generations. However, they are orthogonal and can be naturally combined to shift attention from both token-level and sentence-level, resulting in more effective uncertainty quantification. To achieve that, we simply replace the generative probabilities in Eq. (9), i.e., $p(s_i | x)$ and $p(s_j | x)$, with the token-shifted probability derived from Eq. (8), i.e. $p'(s_i | x) = e^{-\text{TOKEN}SAR(s_i, x)}$ and $p'(s_j | x) = e^{-\text{TOKEN}SAR(s_j, x)}$:

$$E_{T,S}(s_j, S, x) = -\log(p'(s_i | x) + \frac{\sum_{k \neq j} g(s_j, s_k) p'(s_j | x)}{t}). \tag{11}$$

Table 1: Uncertainty estimation AUROCs of TOKEN*SAR*, SENT*SAR*, *SAR*, and baseline methods, across various "off-the-shelf" LLMs and datasets (e.g., CoQA, and Trivia QA). Rouge-L with a threshold of 0.5 is used as the correctness metric.

| Dataset | Model | LS | PE | LN-PE | SE | TOKEN*SAR* | SENT*SAR* | *SAR* |
|---------|-------|-----|-----|-------|-----|-----------|-----------|-------|
| CoQA | OPT-2.7b | 0.531 | 0.692 | 0.706 | 0.699 | 0.707 | 0.717 | **0.735** |
| | OPT-6.7b | 0.542 | 0.696 | 0.723 | 0.717 | 0.724 | 0.722 | **0.750** |
| | OPT-13b | 0.545 | 0.695 | 0.727 | 0.726 | 0.729 | 0.720 | **0.753** |
| | OPT-30b | 0.505 | 0.696 | 0.719 | 0.726 | 0.723 | 0.723 | **0.748** |
| | LLaMA-7b | 0.488 | 0.666 | 0.681 | 0.682 | 0.677 | 0.658 | **0.697** |
| | LLaMA-13b | 0.487 | 0.654 | 0.668 | 0.667 | 0.666 | 0.647 | **0.684** |
| TriviaQA | LLaMA-7b | 0.506 | 0.724 | 0.788 | 0.814 | 0.797 | 0.815 | **0.823** |
| | LLaMA-13b | 0.616 | 0.488 | 0.606 | 0.732 | 0.614 | **0.743** | 0.695 |
| **Average** | | 0.528 | 0.664 | 0.702 | 0.720 | 0.705 | 0.718 | **0.736** |

Then the token- and sentence-level shifted predictive entropy over $K$ generations can be defined as $SAR = \frac{1}{K}\sum_k E_{T,S}(\boldsymbol{s}_k, S, \boldsymbol{x})$.

We denote TOKEN*SAR*, SENT*SAR*, and *SAR* as the token-shifted predictive entropy, sentence-shifted predictive entropy, and both token- and sentence-shifted predictive entropy respectively, in the rest of this paper.

## 5 EMPIRICAL EVALUATIONS

We conduct comprehensive experiments and detailed demographic analyses to evaluate the performance of *SAR* in this section.

### 5.1 EXPERIMENTAL SETTINGS

**Baselines.** We consider 4 baseline methods in our experiments, including Lexical Similarity Lin et al. (2022b), Semantic Entropy (*SE*) Kuhn et al. (2023), Predictive Entropy (*PE*) Kadavath et al. (2022b), and Length-normalized Predictive Entropy (*LN-PE*) Malinin & Gales (2020). Lexical Similarity considers the similarities among multiple generations. *SE* introduces the "semantic equivalence" difficulty in the uncertainty estimation of free-form LLMs and tackles this issue by gathering sentences containing the same meaning into clusters and calculating cluster-wise entropy. *LN-PE* is the length normalized *PE*, i.e., divided by sentence length $N$: $LN\text{-}PE(\boldsymbol{s}, \boldsymbol{x}) = \frac{1}{N}PE(\boldsymbol{s}, \boldsymbol{x})$.

**Models.** We conduct experiments over popular "off-the-shelf" LLMs, including instruction-tuned LLMs (e.g., Vicuna Zheng et al. (2023), LLaMA-2-chat Touvron et al. (2023b), WizardLM Xu et al. (2023)) and pre-trained LLMs (e.g., OPT Zhang et al. (2022) and LLaMA Touvron et al. (2023a)), with model size up to 33B. More details of the used LLMs can be found in A

**Datasets.** We consider 5 free-form question-answering datasets: CoQA Reddy et al. (2019), Trivia QA Joshi et al. (2017), SciQ Welbl et al. (2017), MedQA Jin et al. (2021) and MedMCQA Pal et al. (2022). More details of the used datasets and the splittings can be found in B.

**Correctness Metrics.** We adopt the popular Rouge-L Lin (2004) as the metric when evaluating the correctness of LLMs' generations. Rouge-L deems a generation as correct if its longest common subsequence, regarding ground truth, is larger than a threshold. We set the threshold of Rouge-L as 0.5 by default. We also consider sentence similarity as the correctness metric. We simply deem generations having above 0.5 semantic similarities with the ground truth as correct, measured by SentenceTransformers Reimers & Gurevych (2019a) and use DistillRoBERTa Sanh et al. (2019) as the backbone. We will study the sensitivity of *SAR* to these thresholds in Section 5.4.

**Evaluation Metric.** Following prior work Kuhn et al. (2023), we evaluate uncertainty estimation by predicting the correctness of the model's generations regarding a given question, i.e. to what extent the generated answers can be trusted. The area under the receiver operator characteristic curve (AUROC) indicates the probability that a random correct generation has a lower uncertainty than a random incorrect generation, predicted by uncertainty estimation methods. AUROC equals

Table 2: Uncertainty estimation AUROCs of TOKEN*SAR*, SENT*SAR*, *SAR*, and baseline methods, across various instruction-tuned open-source LLMs, over different datasets (e.g., SciQ, and Trivia QA). The threshold of Rouge-L is set to 0.5. Underline means the second best method.

| Models & Datasets | LS | PE | LN-PE | SE | TOKEN*SAR* ($\Delta SE$) | SENT*SAR* ($\Delta SE$) | *SAR* ($\Delta SE$) |
|---|---|---|---|---|---|---|---|
| **Vicuna-13b** w./ 5 sentences are generated for each question | | | | | | | |
| Trivia QA | 0.560 | 0.690 | 0.624 | 0.630 | 0.692 (+6.2%) | 0.745 (+11.5%) | **0.749** (+11.9%) |
| SciQ | 0.589 | 0.708 | 0.668 | 0.675 | 0.706 (+3.1%) | **0.745** (7.0%) | 0.741 (+6.6%) |
| **Vicuna-33b** w./ 5 sentences are generated for each question | | | | | | | |
| Trivia QA | 0.565 | 0.644 | 0.639 | 0.651 | 0.652 (+0.1%) | **0.715** (+6.4%) | 0.710 (5.9%) |
| SciQ | 0.584 | 0.665 | 0.668 | 0.674 | 0.665 (-0.9%) | **0.717** (+4.3%) | 0.710 (+3.6%) |
| **WizardLM-13b** w./ 5 sentences are generated for each question | | | | | | | |
| Trivia QA | 0.519 | 0.647 | 0.615 | 0.634 | 0.657 (+2.3%) | 0.743 (+10.9%) | **0.744** (+11.0%) |
| SciQ | 0.574 | 0.677 | 0.638 | 0.649 | 0.681 (+3.2%) | **0.719** (+7.0%) | 0.707 (+5.8%) |
| **LLaMA-2-13b-chat** w./ 5 sentences are generated for each question | | | | | | | |
| Trivia QA | 0.504 | 0.647 | 0.615 | 0.622 | 0.654 (+3.2%) | 0.698 (+7.6%) | **0.704** (+8.2%) |
| SciQ | 0.578 | 0.718 | 0.688 | 0.692 | 0.718 (+2.6%) | **0.737** (+4.5%) | 0.725 (+3.3%) |
| **Average** | 0.555 | 0.675 | 0.644 | 0.653 | 0.678 (+2.5%) | **0.727** (+7.4%) | 0.724 (+7.1%) |

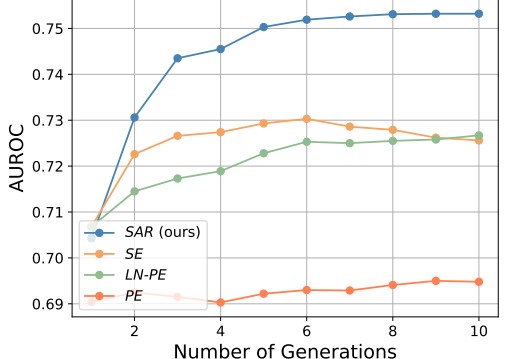

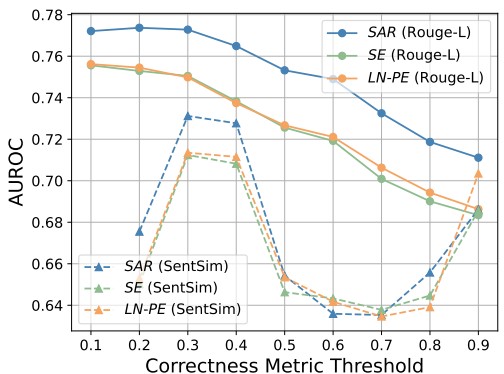

Figure 4: The performance of *SAR* and baseline methods over various numbers of generations. Results are obtained from the OPT-13b model on the CoQA dataset.

Figure 5: The performance of *SAR* over various Rouge-L and Sentence Similarity thresholds. Results are obtained from the OPT-13b model on the CoQA dataset.

0.5 means the assigned uncertainty is no better than random guessing, i.e., they can not differentiate between correct and incorrect generations. AUROC equals 1 means all the correct generations are assigned lower uncertainty than all incorrect generations.

**Hyperparameters.** For OPT-2.7b/6.7b/13b, we generate 10 sentences for each question, i.e. $K$=10. For other models, we generate 5 sentences. The temperature $t$ introduced in Eq. (9) is set to 0.001. We leverage greedy search for all the most likely generations which are used to evaluate correctness, and multinominal sampling for reference generations which are used to estimate uncertainty. More details can be found in Appendix A. All the experiments are conducted on a server with one Intel(R) Xeon(R) Platinum 8358 CPU and two NVIDIA A100 GPUs.

## 5.2 UNCERTAINTY ESTIMATION FOR PRE-TRAINED LLMS

We compare *SAR*, TOKEN*SAR*, and SENT*SAR* with state-of-the-art methods. Results are summarized in Table 1. Generally, our methods significantly outperform prior methods in most of the settings. For instance, *SAR* outperforms other methods by at most 3.6% AUROC over the CoQA dataset, measured by Rouge-L 0.5. The results of setting Rouge-L to 0.3 can be found in Appendix C.4.

Also, the synergy of TOKEN*SAR* and SENT*SAR* achieves remarkable improvements. For instance, TOKEN*SAR* and SENT*SAR* achieve 0.723 AUROC in the OPT-30b-CoQA setting yet combining

them results in 0.748 AUROC. It indicates that TOKEN*SAR* and SENT*SAR* are compatible and can be incorporated effectively.

### 5.3 UNCERTAINTY ESTIMATION FOR INSTRUCTION-TUNED LLMs

We estimate the uncertainty of powerful instruction-tuned LLMs, including Vicuna-13b/33b, LLaMA-2-chat-13b, and WizardLM-13b. All these models are obtained from Huggingface, without any further modifications. Results are summarized in Table 2. It is shown that *S* consistently beat baseline methods in most situations. For example, *SAR* outperforms *SE* by 7.1% AUROC on average, evaluated by Rouge-L 0.5. We also evaluate *SAR* over the AI for science scenarios, such as medical domains.

Table 3: The performance of *SAR* and baseline methods over medical Q&A datasets. Our method achieves better performances for most settings.

| Model | Dataset | *LN-PE* | *SE* | *SAR* |
|---|---|---|---|---|
| Vicuna-13b | MedQA | 0.572 | **0.599** | 0.598 |
| | MedMCQA | 0.649 | 0.685 | **0.717** |
| LLaMA-2-13b-chat | MedQA | 0.562 | 0.609 | **0.616** |
| | MedMCQA | 0.647 | 0.655 | **0.702** |
| WizardLM-13b | MedQA | 0.609 | 0.620 | **0.635** |

As shown in Table 3, we perform experiments over MedQA Jin et al. (2020) and MedMCQA Pal et al. (2022) datasets and our methods achieve better performance for most of the settings. This indicates the potential impacts of our methods on the real world.

### 5.4 ABLATION STUDIES

**Number of Generations.** The effects of the number of generations are summarized in Figure 4. It is shown that our *SAR* is generation-efficient, i.e., it achieves 0.750 AUROC with only 5 generations and it can be consistently boosted with more generations, while other methods may even drop slightly when more generations are provided.

**Sensitivity to Sentence Similarity.** We investigate the sensitivity of SAR to various sentence similarity measurements. Results are reported in Table 4. These models are directly obtained from sentence-transformer ( Appendix D). We show that general-purpose sentence similarity models are more effective than the target LLMs (last column of 4). It is because LLMs are not specifically designed for sentence similarity while these third-party models are designed for this purpose.

Table 4: Sensitivity of *SAR* to sentence similarity measurements. We consider two more models from Sentence-Transformers (Appendix D) and also the target LLMs as the sentence similarity measurement.

| OPT Size | SE | *SAR* w. sentence similarity | | | |
|---|---|---|---|---|---|
| | | RoBERTa | MiniLM | MPNet | OPT-13b |
| 2.7b | 0.699 | **0.735** | 0.723 | 0.723 | 0.716 |
| 6.7b | 0.717 | **0.750** | 0.740 | 0.739 | 0.731 |
| 13b | 0.725 | **0.753** | 0.741 | 0.740 | 0.733 |
| 30b | 0.726 | **0.748** | 0.738 | 0.739 | 0.734 |

**Sensitivity to Correctness Metrics.** The effects of applying different thresholds of correctness metrics are presented in Figure 4. Higher thresholds mean the correctness standards are more harsh. It is shown that the performances of uncertainty quantization will be affected as the metrics are getting harsh. Still, our methods beat baseline methods consistently.

## 6 CONCLUSION

In this paper, we disclose the generative inequality observation in uncertainty estimation: tokens and sentences are created unequally in reflecting semantics yet they are treated equally when estimating uncertainty, which is counter-intuitive. We propose to tackle these inequalities by Shifting Attention to Relevance (*SAR*) from both token-level (TOKEN*SAR*) and sentence-level (SENT*SAR*). Experiments over "off-the-shelf" LLMs demonstrate the superior performances of *SAR*.

**Limitations and Ethics Statement** Our method requires sentence similarity calculations, which might bring additional latency in practice. In addition, our methods require access to token logits. It still might restrict the potential applications of our methods. Our proposed method has the potential to impact the credibility and reliability of LLMs, particularly in the context of reducing hallucination and factual errors.

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

APPENDIX

## A    DETAILS OF LLMS GENERATION

**OPT models.** We will generate 1 most likely generation with the greedy search for all the OPT models. This generation will be used to evaluate the correctness. For OPT-2.7b/6.7b/13b, we will generate 10 sentences for each question with multinomial sampling for uncertainty estimation. For OPT-30b, we will generate 5 sentences. The temperature of generation is fixed at 0.5 for all models. For OPT-2.6b/6.7b/13b, the max length of each generation is set to 256 tokens for the CoQA dataset and SciQ dataset and is set to 128 tokens for the Trivia QA dataset. For OPT-30b, the max length of each generation is set to 128 tokens for all the datasets.

**LLaMA/Vicuna/WizardLM.** We will generate 1 most likely generation with the greedy search and 5 sentences with multinomial sampling for all these models. The max length of each generation is set to 128 tokens. The temperature of generation is set to 0.5.

## B    DATASETS

**CoQA** Reddy et al. (2019) is a large-scale conversational QA task, with more than 127,000 questions. Each question is equipped with a passage to provide contextual information. **Trivia QA** Joshi et al. (2017) is a high-quality reading comprehension dataset that contains over 650k question-answer pairs. These questions are obtained from trivia enthusiasts and answers from Wikipedia. **SciQ** Welbl et al. (2017) dataset is a science-related QA dataset aimed at developing models' capabilities of understanding complex scientific texts. It consists of approximately 13,679 crowdsourced science questions. **MedQA** Jin et al. (2020) is a free-form multiple-choice OpenQA dataset for solving medical problems, collected from the professional medical board exams. **MedMCQA** Pal et al. (2022) is a large-scale, Multiple-Choice Question Answering (MCQA) dataset designed to address real-world medical entrance exam questions.

Following Kuhn et al. (2023), we randomly select around 8,000 questions from the training split of Trivia QA as the questions to be examined. For instruction-tuned experiments, we use 2,000 questions of Trivia QA. We utilize the full validation set (1,000 questions) of SciQ and the development split (7,983 questions) of CoQA. For MedQA and MedMCQA, we also utilize their full validation sets.

## C    ADDITIONAL EXPERIMENTAL ANALYSIS

### C.1    EFFECTS OF *SAR* TEMPERATURE $t$

The hyperparameter $t$ introduced in Eq. (9) is used to control the scale of sentence shifting. The effects of $t$ is provided in Table 5. It is shown that $t$ marginally affects the performance of *SAR*.

Table 5: Effects of temperature $t$ in Eq. (9). Results are evaluated by Rouge-L with 0.5 as the threshold. Results are obtained from *SAR*/TOKEN*SAR*.

| $t$ | OPT-13b | | LLaMA-7b | |
|---|---|---|---|---|
| | CoQA | SciQ | CoQA | TriviaQA |
| $1 \times 10^{-3}$ | 0.753/0.720 | 0.737/0.784 | 0.697/0.658 | 0.823/0.815 |
| $1 \times 10^{0}$ | 0.752/0.719 | 0.739/0.786 | 0.695/0.656 | 0.822/0.816 |
| $1 \times 10^{1}$ | 0.743/0.714 | 0.729/0.786 | 0.686/0.658 | 0.813/0.812 |

### C.2    GENERATION EFFICIENCY

The generation-efficiency of *SAR* on LLaMA-7b-Trivia QA setting is presented in Figure 6.

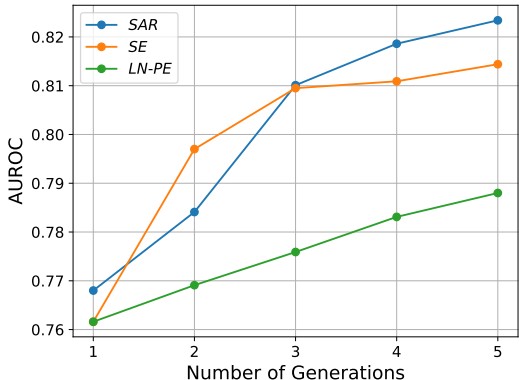

Figure 6: The performance of *SAR* over various numbers of generations. Results are obtained from the LLaMA-7b model over the Trivia QA dataset.

### C.3 SENSITIVITY TO SENTENCE LENGTH.

To study how the *SAR* is affected by sentence length, we quantify the uncertainty rank change for each sentence, caused by *SAR* and SENTß*SAR*. Assume a sentence has a rank of $i$ among all the sentences, evaluated by LN-PE and has a rank of $j$ evaluated by *SAR*, then the uncertainty rank change is $|i - j|$. The correlations between average uncertainty rank change and sentence length are presented in Figure 7. It is shown that our methods tend to conclude medium- and long-length sentences.

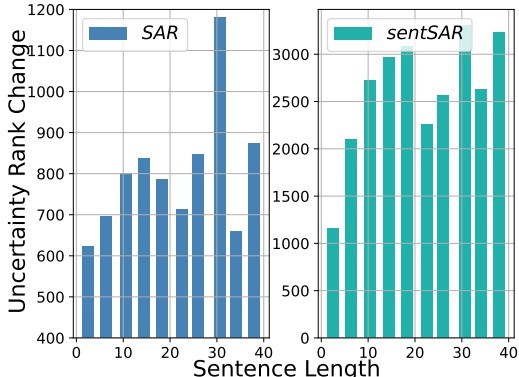

Figure 7: Demographic analysis of sentence length. Uncertainty Rank Change between (**Left**) *SAR* and *LN-PE*, and between (**Right**) SENT*SAR* and *LN-PE*. It is shown that *SAR* and SENT*SAR* are more tend to affect medium- or long-length sentences.

### C.4 DIFFERENT CORRECTNESS METRIC THRESHOLD

We report the results of Rouge-L (0.3) (same as Kuhn et al. (2023) in Table 6.

### C.5 COMPUTATIONAL COSTS ANALYSIS

We would like to highlight that SENT*SAR* is more \*\*generation-efficient\*\*. It surpasses baseline methods under significantly smaller computational constraints. We have quantified the time consumed for each step in the overall uncertainty quantification pipeline. This includes sequence generation, computing logits, semantic clustering for SE, and sentence similarity for SENT*SAR*. We exclude the time taken for aggregating logits/scores as it is negligible (less than 0.001 seconds for all methods). The average time consumed per question, based on an evaluation of 1000 questions from the Vicuna-13b + SciQ dataset, is provided. These measurements were taken using an AMD EPYC 7302 16-Core CPU and a 1xA40 GPU server. Results are summarized in Table 7.

Table 6: Uncertainty estimation AUROCs of TOKEN*SAR*, SENT*SAR*, *SAR*, and baseline methods, across various "off-the-shelf" LLMs and datasets (e.g., CoQA, and Trivia QA). Rouge-L with a threshold of 0.3 is used as the correctness metric.

| Dataset | Model | *LS* | *PE* | *LN-PE* | *SE* | TOKEN*SAR* | SENT*SAR* | *SAR* |
|---|---|---|---|---|---|---|---|---|
| CoQA | OPT-2.7b | 0.573 | 0.666 | 0.719 | 0.712 | 0.719 | 0.689 | **0.742** |
| | OPT-6.7b | 0.588 | 0.671 | 0.745 | 0.741 | 0.746 | 0.696 | **0.768** |
| | OPT-13b | 0.588 | 0.666 | 0.750 | 0.751 | 0.752 | 0.690 | **0.773** |
| | OPT-30b | 0.550 | 0.671 | 0.742 | 0.751 | 0.746 | 0.698 | **0.767** |
| | LLaMA-7b | 0.511 | 0.646 | 0.673 | 0.672 | 0.672 | 0.635 | **0.686** |
| | LLaMA-13b | 0.522 | 0.617 | 0.653 | 0.652 | 0.653 | 0.610 | **0.665** |
| Trivia QA | LLaMA-7b | 0.533 | 0.713 | 0.783 | 0.814 | 0.793 | 0.800 | **0.818** |
| | LLaMA-13b | 0.655 | 0.492 | 0.627 | **0.758** | 0.635 | 0.749 | 0.716 |
| **Average** | | 0.565 | 0.643 | 0.712 | 0.731 | 0.715 | 0.696 | **0.742** |

Table 7: Computational costs of SENT*SAR* and baseline methods. We report both SENT*SAR* with 5 generations and 2 generations.

| Method | Num. of Generations | Generation | Logits Computing | Semantic Clustering | Sentence Similarity | Sum |
|---|---|---|---|---|---|---|
| *PE* | 5 | 4.09s | 1.19s | 0s | 0s | 5.28s |
| *LN-PE* | 5 | 4.09s | 1.19s | 0s | 0s | 5.28s |
| *SE* | 5 | 4.09s | 1.19s | 1.5s | 0s | 6.78s |
| SENT*SAR* | 5 | 4.09s | 1.19s | 0s | 2.58s | 7.86s |
| SENT*SAR* | 2 | 1.64s | 0.48s | 0s | 0.52s | **2.64s** |

Then we compare the 2-generations SENT*SAR* with 5-generations baseline methods. Results are summarized in Table 8. Our SENT*SAR* still surpasses the baseline methods while consuming less than half the time, demonstrating its greater generation efficiency.

Table 8: Comparisons between 2-generations SENT*SAR* and 5-generations baseline methods. SENT*SAR* achieves better performances with only less than half the time consumed by baseline.

| Method | Num. of Generations | Llama-2-13b-chat on SciQ/Trivia QA | Vicuna-13b on SciQ/Trivia QA | Vicuna-33b on SciQ/Trivia QA | WizardLM-13b on SciQ/Trivia QA | Average |
|---|---|---|---|---|---|---|
| *PE* | 5 | **0.718**/0.647 | 0.708/0.690 | 0.665/0.644 | 0.677/0.647 | 0.692/0.657 |
| *LN-PE* | 5 | 0.688/0.615 | 0.668/0.624 | 0.668/0.639 | 0.638/0.615 | 0.666/0.623 |
| *SE* | 5 | 0.692/0.622 | 0.675/0.630 | 0.674/0.651 | 0.649/0.634 | 0.673/0.634 |
| SENT*SAR* | 2 | 0.716/**0.689** | **0.718/0.709** | **0.700/0.674** | **0.697/0.701** | **0.708/0.685** |

# D  SENTENCE SIMILARITY MEASUREMENT

The following is the sentence similarity measurement models we leveraged in Table 4:

- RoBERTa: cross-encoder/stsb-roberta-large
- MiniLM: sentence-transformers/all-MiniLM-L6-v2
- MPNet: sentence-transformers/all-mpnet-base-v2

