# OpenReview forum: "Shifting Attention to Relevance: Towards the Uncertainty Estimation of Large Language Models"
_ICLR.cc/2024/Conference — Submitted to ICLR 2024_

### Official Review · Reviewer_VX9S · 2023-10-31

**Soundness:** 3 good
**Presentation:** 3 good
**Contribution:** 3 good
**Rating:** 6
**Confidence:** 2

**Summary:**

This paper identifies the problem of equal treatment of tokens in LLM-generated text, despite variations in their relevance and representativeness due to linguistic redundancy. Existing methodologies overlook these generative inequalities, leading to biased uncertainty estimation where less semantically significant tokens receive excessive weighting. To rectify this, the proposed method, called Shifting Attention to more Relevant (SAR) components, suggests joint attention shifting at both the token and sentence levels for accurate uncertainty estimation. Extensive experiments are conducted on various LLMs, including instruction-tuned models and pretrained models, using different question-answering tasks across domains like reading comprehension, science, and medical Q&A. The experimental results, along with a demographic analysis, demonstrate SAR's superior performance in addressing the challenges of uncertainty estimation in LLMs. Overall, SAR offers a promising approach to tackle hallucinations and improve uncertainty quantification in the context of LLMs.

**Strengths:**

1.	The authors propose to renormalize the uncertainty scores calculated by tokens with different importance (relevance). The method is intuitive, simple yet pretty effective.

2.	The authors first conduct experiments to verify the existence of generation inequalities, which is a relatively trivial phenomenon. However, the fact that uncertainty estimation is highly affected by the inequalities is also verified and may pose a good contribution to this area.

3.	The experiments are comprehensive and able to prove the effectiveness of the proposed method.

**Weaknesses:**

1.	Although there is stable improvement over the baseline models, it seems marginal.

**Questions:**

1. I have a minor question since I am not an expert in this area: why would the unimportant tokens pose a higher uncertainty? In Figure 1, it seems that the unimportant tokens have relatively less possibility to convey information since the model should have been pretty certain about these tokens in my understanding. However, as shown in Figure 3, this is actually the opposite since the token-level mean of UP is actually higher when the relevance is less. Could you please give an explanation for this?

---

> ### Author Response · Authors · 2023-11-19
> **Official Response to Reviewer VX9S**
>
> **Q1: Although there is stable improvement over the baseline models, it seems marginal.**
>
>
> Thank you for acknowledging our consistent improvements! We would like to present statistics demonstrating that the advancements in SAR are indeed substantial. Our paper includes approximately 21 experiments: 8 on pre-trained Large Language Models (LLMs), 8 on instruction-tuned LLMs, and 5 on medical datasets. We report the enhancements achieved by SAR in comparison to state-of-the-art methods (such as PE, LN-PE, SE), including average improvements, average improvements per setting, and the number of experiments in which SAR outperforms others by at least 2%.
>
> | Method | Total Exp. Number | Avg. Improvement | # Exps with Improvement > 2% | Avg. Pre-trained | Avg. Instruction-tuned | Avg. Medical |
> | ---- | :----: | :----: | :----: | :----: | :----: | :----: |
> | SAR v.s. PE | 16 | 6.1% | 15/16 | 7.2% | 4.9% | - |
> | SAR v.s. LN-PE| 21 | 5.4% | 19/21 | 3.3% | 7.9% | 4.6% |
> | SAR v.s. SE | 21 | 3.7% | 14/21 | 1.5% | 7.1% | 2.0% |
>
>
>
> It is shown that our SAR outperforms state-of-the-art experiments with >2% margins for most experiments. Therefore we would like to mention that the improvement is rather substantial and notable.
>
> **Q2: I have a minor question since I am not an expert in this area: why would the unimportant tokens pose a higher uncertainty? In Figure 1, it seems that the unimportant tokens have relatively less possibility to convey information since the model should have been pretty certain about these tokens in my understanding. However, as shown in Figure 3, this is actually the opposite since the token-level mean of UP is actually higher when the relevance is less. Could you please give an explanation for this?**
>
> R2: Thank you for your insightful comment! ***Why would the unimportant tokens pose a higher uncertainty*** is exactly the question that inspired our method.
>
> We believe this is due to the **intrinsic properties of LLMs**, such as data imbalance in training corpora. For example, if the LLM frequently encounters the phrase "according to" during training, the likelihood of the word "to" following it becomes very high (meaning the uncertainty of "to" is very low) when the preceding token is "according". If the LLM has never encountered "density of" (as shown in Fig. 1 of our paper), the probability of the word "of" following "density" is low (with high uncertainty). Thus, the certainty of LLMs regarding these tokens should be determined by their training. In Figs. 2 and 3, we demonstrate that many irrelevant tokens are assigned lower probabilities by LLMs, which we believe impedes accurate uncertainty estimation.
>
> Our motivation is that important tokens should dominate uncertainty estimation, rather than less significant ones. To achieve this, we quantify the importance of tokens and refocus on these significant tokens during uncertainty quantification, leading to improved performance.
>
>
> We hope these kindly address the reviewer's concerns. We would like to provide responses if the reviewer has further questions.

---

> ### Author Response · Authors · 2023-11-22
> **Have we addressed your concern?**
>
> Dear Reviewer VX9S,
>
> Thank you again for your insightful comments. This is just a gentle reminder as the discussion period ends soon. We would like to know whether we have addressed your concern. Thank you!
>
> Authors of paper 3019

---

### Official Review · Reviewer_UFP5 · 2023-11-01

**Soundness:** 3 good
**Presentation:** 3 good
**Contribution:** 3 good
**Rating:** 6
**Confidence:** 4

**Summary:**

This paper studies uncertainty quantification in long-form generation in language models. While there are many methods for this now, one common method is the predictive entropy, or the log probability of the generated sentence given the prompt. However, predictive entropy assumes all tokens are equally valued when estimating the uncertainty, even though not all of them contribute to the semantics (which determines correctness). To combat this, the paper introduces SAR, which computes a weighted average of the log probability of each added token, weighted by how much each token contributes to the semantics (measured by how much the semantics change when you remove a token). The authors also experiment with a similar procedure for sentences, where the sentence-level uncertainty is measured in terms of similarity to other generated sentences. The authors find that both the token and sentence uncertainties tend to outperform existing baselines, measured by AUROC.

**Strengths:**

* The paper presents intuitive, effective methods for uncertainty estimation; in particular, their method effectively captures the difference between what’s generated (tokens) and what we want to measure uncertainty over (semantics)
* The paper is well-written and easy to follow throughout
* The empirical analysis is good; in particular, their method outperforms baselines across multiple datasets and six different models.

**Weaknesses:**

* The sentSar and tokenSar methods are conceptually quite different, and don’t really make sense to package together (especially since while they are compared, sentSar requires more computation than tokenSar).
* The sentSar method is more computationally expensive than baselines.
* The tables in the paper are very difficult to parse; the font size is small, and presentation could be improved.

**Questions:**

* Is removal the right way to test for change in semantics (rather than just considering likely replacements)?

---

> ### Author Response · Authors · 2023-11-19
> **Official Response to Reviewer UFP5 (1/2)**
>
> **Q1: The sentSar and tokenSar methods are conceptually quite different, and don’t really make sense to package together (especially since while they are compared, sentSar requires more computation than tokenSar).**
>
> Thank you for your insightful comment. We agree that sentSAR and tokenSAR differ conceptually, as they focus on distinct aspects of generation. However, as outlined in Sec. 5.2, they are complementary and can be integrated to address uncertainty quantification effectively at both the token and sentence levels. Consequently, we consider sentSAR and tokenSAR as two separate contributions of our paper. These distinctions are emphasized in our new revision (Sec. 4.3).
>
> **Q2: The sentSar method is more computationally expensive than baselines.**
>
> Thank you for your insightful comment. We agree that sentSAR is more computationally expensive than the baseline methods. However, it's important to highlight that sentSAR is more **generation-efficient**. It surpasses baseline methods under significantly smaller computational constraints. As demonstrated in Fig. 2, SAR, with just 2~3 generations, outperforms all other methods that use 10 generations.
>
> We have quantified the time consumed for each step in the overall uncertainty quantification pipeline. This includes sequence generation, computing logits, semantic clustering for SE, and sentence similarity for sentSAR. We exclude the time taken for aggregating logits/scores as it is negligible (less than 0.001 second for all methods). The average time consumed per question, based on an evaluation of 1000 questions from the Vicuna-13b + SciQ dataset, is provided. These measurements were taken using an AMD EPYC 7302 16-Core CPU and a 1xA40 GPU server:
>
> | Method | Number of Generations | Generation | Logits Computing | Semantic Clustering | Sentence Similarity |  sum |
> | ---- | :----: | :----: | :----: | :----: |  :----: | :----: |
> | PE | 5|4.09s|	1.19s|	0s|	0s|	5.28s|
> | LN-PE | 5| 4.09s|	1.19s|	0s|	0s|	5.28s|
> | SE | 5|4.09s|	1.19s|	1.5s|	0s|	6.78s|
> | sentSAR | 5|4.09s|	1.19s|	0s|	2.58s|	7.86s|
> | sentSAR | 2 |	1.64s|	0.48s|	0s|	0.52s|	**2.64s**|
>
> We present two scenarios for our sentSAR: 1) It takes **7.86 seconds** when utilizing 5 generations; 2) The time reduces to **2.64 seconds** when using just **2 generations**, which is the minimum number of generations required for sentSAR.
>
> Then we compare the 2-generations sentSAR with 5-generations baseline methods:
>
> | Method | Number of Generations | Llama-2-13b-chat + SciQ/Trivia QA | Vicuna-13b + SciQ/Trivia QA| Vicuna-33b + SciQ/Trivia QA|WizardLM-13b + SciQ/Trivia QA|average|
> | ---- | :----: | :----: | :----: | :----: | :----: |:----:|
> |PE|5|**0.718**/0.647|0.708/0.690|0.665/0.644|0.677/0.647|0.692/0.657|
> |LN-PE|5|0.688/0.615|0.668/0.624|0.668/0.639|0.638/0.615|0.666/0.623|
> |SE|5|0.692/0.622|0.675/0.630|0.674/0.651|0.649/0.634|0.673/0.634|
> |sentSAR|**2**|0.716/**0.689**|**0.718**/**0.709**|**0.700**/**0.674**|**0.697**/**0.701**|**0.708**/**0.685**|
>
> Our sentSAR still surpasses the baseline methods while consuming less than half the time, demonstrating its greater generation efficiency. We have included these results in the updated version of our paper (Appendix C.5).

---

> ### Author Response · Authors · 2023-11-19
> **Official Response to Reviewer UFP5 (2/2)**
>
> **Q3: The tables in the paper are very difficult to parse; the font size is small, and presentation could be improved.**
>
> Thank you for pointing this out! We have modified the formats and font sizes of tables in our new revision.
>
> **Q4: Is removal the right way to test for change in semantics (rather than just considering likely replacements)?**
>
> Thank you for your insightful comment! We believe that removal is an appropriate approach for our method for the following reasons:
>
> 1. Removal allows for the quantification of per-token relevance and is easy to implement.
> 2. Removal strategies are widely employed in existing works[1-2].
> 3. The grammar mistakes or fluency issues caused by removal are not significant for current sentence similarity measurements. For example, consider the following SciQ example, which demonstrates the changes in similarity after the removal of each token, as measured by the cross-encoder-roberta-large model:
>
> | | Mesophilic | has |important |uses| in |food| preparation| including| cheese| yogurt| beer| and |wine|
> | ---- | :----:| :----: |:----: |:----: |:----:| :----: |:----:| :----:| :----: |:----: |:----:| :----: | :----: |
> | $\Delta$ | 0.25 | 0.01 | 0.02 | 0.05 | 0.02 | 0.10 | 0.04 | 0.01 | 0.16 | 0.19 | 0.21 | 0.01 | 0.19 |
>
> The similarity significantly changes only when key tokens (such as Mesophilic, food, cheese, yogurt, beer, wine) are removed. This is conceptually aligned with human understanding and also meets our requirements effectively.
>
> Using replacements could be less effective because if we substitute a token with synonyms, the semantic impact is minimal, making it challenging to quantify token relevance accurately. Additionally, finding suitable replacements for some words or tokens, like definite and indefinite articles, prepositions, etc., can be difficult. We are open to conducting evaluations if the reviewer can suggest specific strategies for replacements.
>
> Reference:
>
> [1] Lee, Joosung. "Stable style transformer: Delete and generate approach with encoder-decoder for text style transfer." arXiv preprint arXiv:2005.12086 (2020).
>
> [2] Liu, Junyi, et al. "TCRA-LLM: Token Compression Retrieval Augmented Large Language Model for Inference Cost Reduction." arXiv preprint arXiv:2310.15556 (2023).
>
>
> We hope these kindly address the reviewer's concerns. We would like to provide responses if the reviewer has further questions.

---

> ### Author Response · Authors · 2023-11-22
> **Have we addressed your concern?**
>
> Dear Reviewer UFP5,
>
> Thank you again for your insightful comments. This is just a gentle reminder as the discussion period ends soon. We would like to know whether we have addressed your concern. Thank you!
>
> Authors of paper 3019

---

### Official Review · Reviewer_i37c · 2023-11-02

**Soundness:** 3 good
**Presentation:** 3 good
**Contribution:** 3 good
**Rating:** 6
**Confidence:** 4

**Summary:**

The authors present a new method for quantifying uncertainty in LLM outputs that involves weighting token probabilities by an estimate of the token's relative importance ("relevance"). This is combined with a second sentence-level uncertainty measure which adjusts a sentence's importance measure according to how similar it is to other generated sentences. When combined these uncertainty measures outperform multiple baselines across various models, and datasets, and evaluation methods.

**Strengths:**

The presented uncertainty estimation is built up out of simple to understand pieces, and yield impressive empirical results compared to competing techniques.

**Weaknesses:**

I believe there is a typo in Equation 9. According to the definition of R_S given in Equation 4, the sum in the second line should be over $g(s_j,s_k)p(s_k|x)$ not $g(s_j,s_k)p(s_j|x)$.

"The reason we normalize relevance score in Eq. (6) is two-fold: a) to make tokens comparable within a sentence" - this doesn't make sense, the relative token relevances are the same before and after the normalization, they're all being scaled by the same factor

Table 1 is very crowded (multiple models, multiple datasets, multiple Rouge-L cutoffs, multiple baselines, multiple SAR methods). Maybe stick with just 1 RL cutoff and if you want to include the other cutoff, put it in the appendix.

**Questions:**

“The area under the receiver operator characteristic curve (AUROC) metric is equivalent to the probability that a randomly chosen correct answer has a higher uncertainty score than a randomly chosen incorrect answer.” - Shouldn’t this be the opposite? If the model’s uncertainty is high, then it implies that the model is more likely to get the answer wrong, no? I realize that this was copied from Kuhn et al, but this still seems like an important point to understand.

---

> ### Author Response · Authors · 2023-11-19
> **Response to Reviewer i37c**
>
> **Q1: I believe there is a typo in Equation 9. According to the definition of R_S given in Equation 4, the sum in the second line should be over $g(s_j, s_k)p(s_k|x)$ not $g(s_j, s_k)p(s_j|x)$.**
>
> R1: Thank you for pointing this out! It is a typo. We have corrected this in our new revision.
>
> **Q2: "The reason we normalize relevance score in Eq. (6) is two-fold: a) to make tokens comparable within a sentence" - this doesn't make sense, the relative token relevances are the same before and after the normalization, they're all being scaled by the same factor**
>
> R2: Thank you for pointing this out! You are right, this is a typo. We mean with normalization, token relevances are comparable **across** sentences. We have fixed this statement in our new revision.
>
> **Q3: Table 1 is very crowded (multiple models, multiple datasets, multiple Rouge-L cutoffs, multiple baselines, multiple SAR methods). Maybe stick with just 1 RL cutoff and if you want to include the other cutoff, put it in the appendix.**
>
> R3: Thank you for your suggestion! We have split Table 1 into two parts and only keep the results of RL-0.5 in the main body of our new revision. The results of RL-0.3 are placed in Appendix C.4.
>
> **Q4: "The area under the receiver operator characteristic curve (AUROC) metric is equivalent to the probability that a randomly chosen correct answer has a higher uncertainty score than a randomly chosen incorrect answer." - Shouldn’t this be the opposite? If the model’s uncertainty is high, then it implies that the model is more likely to get the answer wrong, no?**
>
> R4: Thank you for pointing this out! We indeed follow the setting from the SE paper and we didn’t realize this typo before. We have corrected this description as the following (also in our new revision):
>
> Following prior work (Kuhn et al.), we evaluate uncertainty estimation by predicting the correctness of the model's generations regarding a given question, i.e. to what extent the generated answers can be trusted. The area under the receiver operator characteristic curve (AUROC) indicates the probability that a random correct generation has a lower uncertainty than a random incorrect generation, predicted by uncertainty estimation methods. AUROC equals 0.5 means the assigned uncertainty is no better than random guessing, i.e., they can not differentiate between correct and incorrect generations. AUROC equals 1 means all the correct generations are assigned lower uncertainty than all incorrect generations.
>
> We hope these kindly address the reviewer's concerns. We would like to provide responses if the reviewer has further questions.

---

> ### Author Response · Authors · 2023-11-22
> **Have we addressed your concern?**
>
> Dear Reviewer i37c,
>
> Thank you again for your insightful comments. This is just a gentle reminder as the discussion period ends soon. We would like to know whether we have addressed your concern. Thank you!
>
> Authors of paper 3019

---

### Author Response · Authors · 2023-11-19
**General Response**

Thank you for the positive feedback from all the reviewers and for their constructive suggestions. We have updated our manuscript with the following modifications:

- **Section 4.3**: We highlighted that tokenSAR and sentSAR though conceptually different are orthogonal and compatible, which can be seen as two individual contributions of our paper.
- **Section 5.1**: We updated the descriptions of the evaluation metric and made the uncertainty ranks clear.
- **Table1**: We split Table 1 into two parts (results of RL-0.5 and results of RL-0.3). We only keep the results of RL-0.5 in the main body and place RL-0.3 in Appendix C.4, which makes the comparisons much easier and clearer.
- **Table 2/3/4**: We modified the formats and styles of these tables to make them easy to read.
- **Appendix C.5**: We added the computational cost comparisons between our method and baseline methods, highlighting our method is generation-efficient, i.e., achieves better performances with less than half the time consumed by baseline methods.

The details of revisions can be found in our official responses and revised paper. Please do not hesitate to contact us for any further suggestions or discussions.

Authors of paper 3019

---

### Meta-Review · Area_Chair_reA2 · 2023-12-06

**Metareview:**

The paper quantifies the uncertain of LLM generation, which may help downstream tasks such as detecting hallucination. Specifically, the proposed metric has two variants: sentSar and tokenSar, where the main contribution is to include a weighting coefficient for certain words.

Reviewers find the paper borderline. While the approach is intuitive, it gives marginal performance improvement. My main concern is that the proposed method is a pure heuristic (and arguably with better engineering the performance may be further improved).

**Justification For Why Not Higher Score:**

The proposed metric is a pure heuristic that lacks principles and yields marginal performance improvement.

**Justification For Why Not Lower Score:**

N/A

---

### Decision · Program_Chairs · 2024-01-16

Reject